Data-based intervention approach for Complexity-Causality measure

Kathpalia Aditi kathpaliaaditi@gmail.com
Nagaraj Nithin
Consciousness Studies Programme, National Institute of Advanced Studies , Bengaluru , Karnataka , India
Cattuto Ciro
Electronic publication date: 2019 May 27
Publication date: 2019
Volume: 5
Electronic Location ID: e196
Received 2018 Dec 6; Accepted 2019 Apr 29
Copyright: ©2019 Kathpalia et al.
Copyright year: 2019
Copyright holder: Kathpalia et al.
License: This is an open access article distributed under the terms of the Creative Commons Attribution License, which permits unrestricted use, distribution, reproduction and adaptation in any medium and for any purpose provided that it is properly attributed. For attribution, the original author(s), title, publication source (PeerJ Computer Science) and either DOI or URL of the article must be cited.
License URL: https://creativecommons.org/licenses/by/4.0/

Keywords: Causality, Causal inference, Intervention, Compression-complexity, Model-based, Dynamical complexity, Negative causality

Funding: Tata Trusts and Cognitive Science Research Initiative (CSRI-DST) Grant No. DST/CSRI/2017/54 This work was supported by Tata Trusts and Cognitive Science Research Initiative (CSRI-DST) Grant No. DST/CSRI/2017/54. The funders had no role in study design, data collection and analysis, decision to publish, or preparation of the manuscript.

==============================
Causality testing methods are being widely used in various disciplines of science. Model-free methods for causality estimation are very useful, as the underlying model generating the data is often unknown. However, existing model-free/data-driven measures assume separability of cause and effect at the level of individual samples of measurements and unlike model-based methods do not perform any intervention to learn causal relationships. These measures can thus only capture causality which is by the associational occurrence of ‘cause’ and ‘effect’ between well separated samples. In real-world processes, often ‘cause’ and ‘effect’ are inherently inseparable or become inseparable in the acquired measurements. We propose a novel measure that uses an adaptive interventional scheme to capture causality which is not merely associational. The scheme is based on characterizing complexities associated with the dynamical evolution of processes on short windows of measurements. The formulated measure, Compression-Complexity Causality is rigorously tested on simulated and real datasets and its performance is compared with that of existing measures such as Granger Causality and Transfer Entropy. The proposed measure is robust to the presence of noise, long-term memory, filtering and decimation, low temporal resolution (including aliasing), non-uniform sampling, finite length signals and presence of common driving variables. Our measure outperforms existing state-of-the-art measures, establishing itself as an effective tool for causality testing in real world applications.

Introduction

The ‘Ladder of Causation’ very rightly arranges hierarchically the abilities of a causal learner (Pearl & Mackenzie, 2018). The three levels proposed are: 1. Association, 2. Intervention and 3. Counterfactuals, when arranged from the lower rung to the higher rung. Currently, causality learning and inferring algorithms using only data are still stuck at the lowermost rung of ‘Association’.

Measures such as Granger Causality (GC) (Granger, 1969) and its various modifications (Dhamala, Rangarajan & Ding, 2008; Marinazzo, Pellicoro & Stramaglia, 2008), as well as, Transfer Entropy (TE) (Schreiber, 2000) that are widely being used across various disciplines of science—neuroscience (Seth, Barrett & Barnett, 2015; Vicente et al., 2011), climatology (Stips et al., 2016; Mosedale et al., 2006), econometrics (Hiemstra & Jones, 1994; Chiou-Wei, Chen & Zhu, 2008), engineering (Bauer et al., 2007) etc. are largely ‘model-free’/ ‘data-driven’ measures of causality. They make minimal assumptions about the underlying physical mechanisms and depend more on time series characteristics (Seth, Barrett & Barnett, 2015). Hence, they have a wider scope compared to specific model assumptions made by methods such as Dynamic Causal Modelling (Friston, Harrison & Penny, 2003) and Structural Equation Modeling (Pearl, 2009). However, the assumptions made by these methods are often ignored in practice, resulting in erroneous causality estimates on real world datasets. These measures can accurately quantify the degree of coupling between given time series only if assumptions (such as linearity, stationarity and presence of Gaussian noise in case of GC and stationarity, markovian in case of TE) are satisfied. Thus, these methods, when correctly applied, can infer the presence of causality when it is by ‘association’ alone and not due to higher levels on the Ladder of Causation. To explain this better, consider a case where the ‘cause’ and ‘effect’ are inseparable. This can happen even when the time series satisfies stationarity but is non-markovian or in several instances when it is non-stationary. In fact, the stated assumptions are quite unlikely to be met in practice considering that acquired data are typically samples of continuous/discrete evolution of real world processes. These processes might be evolving at spatio-temporal scales very different from the scales of measurements. As a result, cause and effect may co-exist in a single measurement or overlap over blocks of measurements, making them inseparable. In such a scenario, it would be incorrect to estimate causality by means of correlations and/or joint probabilities which implicitly assumes the separability of ‘cause’ and ‘effect’. Both GC and TE make this assumption of separability. Circularly, to characterize a time series sample as purely a ‘cause’ or an ‘effect’ is possible only if there is a clear linear/markovian separable relationship. When cause and effect are inseparable, ‘associational’ measures of causality such as GC and TE are insufficient and we need a method to climb up the ladder of causation.

Intervention based approaches to causality rank higher than association. It involves not just observing regularities in the data but actively changing what is there and then observing its effect. In other words, we are asking the question—what will happen if we ‘do’ something? Given only data and not the power to intervene on the experimental set up, intervention can only be done by building strong, accurate models. Model-based causality testing measures, alluded to before, will fall in this category. They invert the model to obtain its various parameters, and then intervene to make predictions about situations for which data is unavailable. However, these methods are very domain specific and the models require specific knowledge about the data. With insufficient knowledge about the underlying model which generated the data, such methods are inapplicable.

Given only data that has already been acquired without any knowledge of its generating model or the power to intervene on the experimental/real-world setting, we can ask the question—what kind of intervention is possible (if at all) to infer causality? The proposed ‘interventional causality’ approach will not merely measure ‘associational causality’ because it does not make the assumption that the cause and its effect are present sample by sample (separable) as is done by existing model-free, data based methods of causality estimation.

Even in cases where cause and its effect are inseparable, which is probably true for most real-world processes, the change in the dynamics of processes would contain information about causal influences between them. With this understanding, we propose the novel idea of data-based, model-free Interventional Complexity Causality (ICC). In this paper, we formalize the notion of ICC using Compression-Complexity to define Compression-Complexity Causality (CCC). CCC shows some interesting properties. We test CCC on simulated and real datasets and compare its performance with existing model-free causality methods. Our results demonstrate that CCC overcomes the limitations of ‘associational’ measures (GC and TE) to a large extent.

Other methods for causality estimation based on compression have been proposed in literature (Budhathoki & Vreeken, 2016; Wieczorek & Roth, 2016), but the very philosophy behind our method and its implementation are very different from these existing methods.

This paper is organized as follows. The idea of Dynamical Complexity and its specific realization Dynamical Compression-Complexity are discussed in ‘Dynamical Complexity (DC) and Dynamical Compression-Complexity (CC)’. Interventional Complexity Causality and its specific case Compression-Complexity Causality (CCC) are discussed in ‘Interventional Complexity Causality (ICC) and Compression-Complexity Causality (CCC)’. How it is possible to obtain positive and negative values of CCC and what its implications are on the kind of causal influence is detailed in ‘Positive and Negative CCC’. Results and discussion on the performance of CCC and its comparison with existing measures, GC and TE, are included in ‘Results and Discussion’. This is followed by conclusions and future work in ‘Conclusions’. A list of frequently used abbreviations is provided at the end of the paper.

Dynamical Complexity (DC) and Dynamical Compression-Complexity (CC)

There can be scenarios where cause and effect co-exist in a single temporal measurement or blocks of measurements. For example, this can happen (a) inherently in the dynamics of the generated process, (b) when cause and effect occur at different spatio-temporal scales, (c) when measurements are acquired at a scale different from the spatio-temporal scale of the cause–effect dynamics (continuous or discrete). In such a case, probabilities of joint occurrence is too simplistic an assumption to capture causal influences. On the other hand, the very existence of causality here is actually resulting in a change of joint probabilities/correlations which cannot be captured by an assumption of static probabilities. To overcome this problem, we capture causality using the idea of dynamical complexity. Inseparable causal influences within a time series (or between two time series) would be reflected in their dynamical evolution. Dynamical Complexity (DC) of a single time series X is defined as below - (1) DCΔX|Xpast=CXpast+ΔX−CXpast,

where ΔX is a moving window of length w samples and Xpast is a window consisting of immediate past L samples of ΔX. ‘+’ refers to appending, for e.g., for time series A = [1, 2, 3] and B = [p, q], then A + B = [1, 2, 3, p, q].C(X) refers to complexity of time series X.DC, thus varies with the temporal index of ΔX and can be averaged over the entire time series to estimate its average DC.

It is important to note that dynamical complexity is very different from complexity rate (CR), which can be estimated as follows - (2) CRΔX|Xpast=CXpast,ΔX−CXpast,

where C(Xpast, ΔX) is the joint complexity of Xpast and ΔX. Complexity rate can be seen as a generalization of Shannon entropy rate (Cover & Thomas, 2012), the difference being that the former can be computed using any notion of complexity, not just entropy. As is evident from Eqs. (1) and (2), CR is estimated based on the joint occurrences of ΔX and Xpast, while DC captures temporal change in complexity on the evolution of the process. In case of the inseparability of cause and effect, it would be inappropriate to use CR to infer causal relationships.

Now for this notion of “complexity”, that has been referred to in this section several times, there is no single unique definition. As noted in Nagaraj & Balasubramanian (2017b), Shannon entropy (Shannon, 1948; Cover & Thomas, 2012) is a very popular and intuitive measure of complexity. A low value of Shannon entropy indicates high redundancy and structure (low complexity) in the data and a high value indicates low redundancy and high randomness (high complexity). For ergodic sources, owing to Shannon’s noiseless source coding theorem (Cover & Thomas, 2012), (lossless) compressibility of the data is directly related to Shannon entropy. However, robustly estimating compressibility using Shannon entropy for short and noisy time series is a challenge (Nagaraj & Balasubramanian, 2017a). Recently, the notion of compression-complexity has been introduced (Nagaraj & Balasubramanian, 2017a) to circumvent this problem. Compression-complexity defines the complexity of a time series by using optimal lossless data compression algorithms. It is well acknowledged that data compression algorithms are not only useful for compression of data for efficient transmission and storage, but also act as models for learning and statistical inference (Cilibrasi, 2007). Lempel–Ziv (LZ) Complexity (Lempel & Ziv, 1976) and Effort-To-Compress (ETC) (Nagaraj, Balasubramanian & Dey, 2013) are two measures which fall in this category.

As per the minimum description length principle (Rissanen, 1978), that formalizes the Occam’s razor, the best hypothesis (model and its parameters) for a given set of data is the one that leads to its best compression. Extending this principle for causality, an estimation based on dynamical complexity (compressibility) of time series would be the best possible means to capture causally influenced dynamics.

Out of the complexity measures discussed before, ETC seemed to be most suitable for estimation of dynamical complexity. ETC is defined as the effort to compress the input sequence using the lossless compression algorithm known as Non-sequential Recursive Pair Substitution (NSRPS). It has been demonstrated that both LZ and ETC outperform Shannon entropy in accurately characterizing the dynamical complexity of both stochastic (Markov) and deterministic chaotic systems in the presence of noise (Nagaraj & Balasubramanian, 2017a; Nagaraj & Balasubramanian, 2017b). Further, ETC has shown to reliably capture complexity of very short time series where even LZ fails (Nagaraj & Balasubramanian, 2017a), and for analyzing short RR tachograms from healthy young and old subjects (Balasubramanian & Nagaraj, 2016). Recently, ETC has been used to propose a compression-complexity measure for networks (Virmani & Nagaraj, 2019).

In order to faithfully capture the process dynamics, DC is required to be estimated on overlapping short-length windows of time series data. Infotheoretic quantities (like shannon entropy), which are based on the computation of probability densities, are not the ideal choice here (owing to finite-length effects). Compression-Complexity measures are more appropriate choices. Because of the advantages of ETC over LZ mentioned above, we use ETC to formulate our measure of causality discussed in the next section. Before that, we describe how individual and joint compression complexities are computed using ETC (Nagaraj, Balasubramanian & Dey, 2013) in the subsections below.

ETC measure for a time series: ETC(X)

Since ETC expects a symbolic sequence as its input (of length >1), the given time series should be binned appropriately to generate such a sequence. Once such a symbolic sequence is available, ETC proceeds by parsing the entire sequence (from left to right) to find that pair of symbols in the sequence which has the highest frequency of occurrence. This pair is replaced with a new symbol to create a new symbolic sequence (of shorter length). This procedure is repeated iteratively and terminates only when we end up with a constant sequence (whose entropy is zero since it consists of only one symbol). Since the length of the output sequence at every iteration decreases, the algorithm will surely halt. The number of iterations needed to convert the input sequence to a constant sequence is defined as the value of ETC complexity. For example, the input sequence ‘12121112’ gets transformed as follows: 12121112↦33113↦4113↦513↦63↦7. Thus, ETC(12121112) = 5. ETC achieves its minimum value (0) for a constant sequence and maximum value (m − 1) for a m length sequence with distinct symbols. Thus, we normalize the ETC complexity value by dividing by m − 1. Thus, normalized ETC12121112=57. Note that normalized ETC values are always between 0 and 1 with low values indicating low complexity and high values indicating high complexity.

Joint ETC measure for a pair of time series: ETC(X, Y)

We perform a straightforward extension of the above mentioned procedure (ETC(X)) for computing the joint ETC measure ETC(X, Y) for a pair of input time series X and Y of the same length. At every iteration, the algorithm scans (from left to right) simultaneously X and Y sequences and replaces the most frequent jointly occurring pair with a new symbol for both the pairs. To illustrate it by an example, consider, X = 121212 and Y = abacac. The pair (X, Y) gets transformed as follows: (121212, abacac)↦(1233, abdd)↦(433, edd)↦(53, fd)↦(6, g). Thus, ETC(X, Y) = 4 and normalized value is 45. It can be noted that ETC(X, Y) ≤ ETC(X) + ETC(Y).

Interventional Complexity Causality (ICC) and Compression-Complexity Causality (CCC)

To measure how the dynamics of a process Y influence the dynamics of a process X, we intervene to create new hypothetical blocks of time series data, Ypast + ΔX, where Ypast is a window of length L samples, taken from the immediate past of the window ΔX. These blocks are created by ‘surgery’ and do not exist in reality in the data that is already collected. Interventional Complexity Causality (ICC) is defined as the change in the dynamical complexity of time series X when ΔX is seen to be generated jointly by the dynamical evolution of both Ypast and Xpast as opposed to by the reality of the dynamical evolution of Xpast alone.

This formulation is actually in line with Wiener’s idea, according to which, time series Y causes X, if incorporating the past of Y helps to improve the prediction of X (Wiener, 1956). While GC is based on the notion of improved predictability and TE on the reduction of uncertainty, ICC is based on the notion of change in ‘dynamical complexity’ when information from the past of Y is brought in, in order to check its causal influence on X. The difference between existing approaches and the proposed measure is that the effect of Y on X is analyzed based on ‘associational’ means in case of the former and by ‘interventional’ means in case of the latter. With this formulation, ICC is designed to measure effect, like GC and TE, and not the mechanism, as in Dynamic Causal Modelling (Seth, Barrett & Barnett, 2015; Barrett & Barnett, 2013). To elaborate on this aspect, ICC cannot explicitly quantify the interaction coefficients of the underlying generative model (physical mechanism), but will only estimate causal influence based on change in dynamical complexities. It is, however, expected that ICC will be closer to the underlying mechanism than existing methods, because, by its very formulation, it taps on causes and their effects based on dynamical evolution of processes.

Mathematically, (3) ICCYpast→ΔX=DCΔX|Xpast−DCΔX|Xpast,Ypast,

where DC(ΔX|Xpast) is as defined in Eq. (1) and DC(ΔX|Xpast, Ypast) is as elaborated below: (4) DCΔX|Xpast,Ypast=CXpast+ΔX,Ypast+ΔX−CXpast,Ypast,

where C(⋅, ⋅) refers to joint complexity. ICC varies with the moving temporal window ΔX and its corresponding Ypast, Xpast. To estimate average causality from time series Y to X, ICCYpast→ΔX obtained for all ΔXs are averaged.

The above is the generic description of ICC that can be estimated using any complexity measure. For the reasons discussed in ‘Dynamical Complexity (DC) and Dynamical Compression-Complexity (CC)’, we would like to estimate ICC using the notion of Dynamical Compression-Complexity estimated by the measure ETC. The measure would then become Interventional Compression-Complexity Causality. For succinctness, we refer to it as Compression-Complexity Causality (CCC). To estimate CCC, time series blocks Xpast, Ypast, Xpast + ΔX, and surgically created Ypast + ΔX are separately encoded (binned)—converted to a sequence of symbols using ‘B’ uniformly sized bins for the application of ETC.1 For the binned time series blocks, Xpast, Ypast, Xpast + ΔX, Ypast + ΔX, to determine whether Ypast caused ΔX or not, we first compute dynamical compression-complexities, denoted by CC, (5) CCΔX|Xpast=ETCXpast+ΔX−ETCXpast,

(6) CCΔX|Xpast,Ypast=ETCXpast+ΔX,Ypast+ΔX−ETCXpast,Ypast,

Equation (5) gives the dynamical compression-complexity of ΔX as a dynamical evolution of Xpast alone. Equation (6) gives the dynamical compression-complexity for ΔX as a dynamical evolution of both Xpast and Ypast. ETC(⋅) and ETC(⋅, ⋅) refer to individual and joint effort-to-compress complexities. For estimating ETC from these small blocks of data, short-term stationarity of X and Y is assumed.

We now define Compression-Complexity Causality CCCYpast→ΔX as: (7) CCCYpast→ΔX=CCΔX|Xpast−CCΔX|Xpast,Ypast.

Averaged CCC from Y to X over the entire length of time series with the window ΔX being slided by a step-size of δ is estimated as— (8) CCCY→X=CCC¯Ypast→ΔX=CC¯ΔX|Xpast−CC¯ΔX|Xpast,Ypast,

If CC¯ΔX|Xpast,Ypast≈CC¯ΔX|Xpast, then CCCY→X is statistically zero, implying no causal influence from Y to X. If CCCY→X is statistically significantly different from zero, then we infer that Y causes X. A higher magnitude of CCCY→X implies a higher degree of causation from Y to X. The length of Xpast, Ypast, that is L is chosen by determining the correct intervention point. This is the temporal scale at which Y has a dynamical influence on X. Detailed criteria and rationale for estimating L and other parameters used in CCC estimation: w (length of ΔX), δ and B for any given pair of time series are discussed in Section S3. CCC is invariant to local/global scaling and addition of constant value to the time series. As CCC is based on binning of small blocks of time series data, it is noise resistant. Furthermore, it is applicable to non-linear and short term stationary time series. Being based on dynamical evolution of patterns in the data, it is expected to be robust to sub-sampling and filtering.

For multivariate data, CCC can be estimated in a similar way by building dictionaries that encode information from all variables. Thus, to check conditional causality from Y to X amidst the presence of other variables (say Z and W), two time varying dictionaries are built—D that encodes information from all variables (X, Y, Z, W) and D′ that encodes information from all variables except Y (X, Z, W only). Once synchronous time series blocks from each variable are binned, the dictionary at that time point is constructed by obtaining a new sequence of symbols, with each possible combination of symbols from all variables being replaced by a particular symbol. The mechanism for construction of these dictionaries is discussed in Section S1. Subsequently, dynamical compression-complexities are computed as: (9) CCΔX|Dpast′=ETCDpast′+ΔX−ETCDpast′,

(10) CCΔX|Dpast=ETCDpast+ΔX−ETCDpast,

where Dpast′+ΔX represents the lossless encoding of joint occurrences of binned time series blocks Xpast + ΔX, Zpast + ΔX, Wpast + ΔX and Dpast′ refers to the lossless encoding of joint occurrences of binned time series blocks Xpast, Zpast and Wpast. Similarly, Dpast + ΔX represents the lossless encoding of joint occurrences of binned time series blocks Xpast + ΔX, Ypast + ΔX, Zpast + ΔX, Wpast + ΔX and Dpast refers to the the lossless encoding of joint occurrences of binned time series blocks Xpast, Ypast, Zpast and Wpast.

Conditional Compression-Complexity Causality, CCCYpast→ΔX|Zpast,Wpast, is then estimated as the difference of Eqs. (9) and (10). Averaged Conditional Compression Complexity-Causality over the entire time series with the window ΔX being slided by a step-size of δ is given as below: (11) CCCY→X|Z,W=CC¯ΔX|D′−CC¯ΔX|D.

Positive and negative CCC

The dynamical compression-complexities estimated for the purpose of CCC estimation, CC(ΔX|Xpast) and CC(ΔX|Xpast, Ypast), can be either positive or negative. For instance, consider the case when CC(ΔX|Xpast) becomes negative. This happens when ETC(Xpast + ΔX) is less than ETC(Xpast), which means that with the appending of ΔX, the sequence Xpast has become more structured resulting in reduction of its complexity. The value of CC(ΔX|Xpast) is positive when appending of ΔX makes Xpast less structured (hence more complex). Similarly, CC(ΔX|Xpast, Ypast) can also become negative when ETC realizes Xpast + ΔX, Ypast + ΔX to be more structured than Xpast, Ypast. When the opposite is true, CC(ΔX|Xpast, Ypast) is positive.

Because of the values that CC(ΔX|Xpast) and CC(ΔX|Xpast, Ypast) can take, CCCYpast→ΔX can be both positive or negative. How different cases result with different signs of the two quantities along with their implication on CCC is shown in Table S1 of the supplementary material. We see that the sign of CCCYpast→ΔX signifies the ‘kind of dynamical influence’ that Ypast has on ΔX, whether this dynamical influence is similar to or different from that of Xpast on ΔX. When CCCYpast→ΔX is −ve, it signifies that Ypast has a different dynamical influence on ΔX than Xpast. On the contrary, when CCCYpast→ΔX is +ve, it signifies that Ypast has a dynamical influence on ΔX that is similar to that of Xpast. On estimating the averaged CCC from time series Y to X, expecting that CCCYpast→ΔX values do not vary much with time, we can talk about the kind of dynamical influence that time series Y has on X. For weak sense stationary processes, it is intuitive that the influence of Y on X would be very different from that on X due to its own past when the distributions of coupled time series Y and X are very different.

We verify this intuition by measuring probability distribution distances2 between coupled processes Y and X using symmetric Kullback–Leibler Divergence (KL) and Jensen–Shannon Divergence (JSD). The trend of values obtained by these divergence measures is compared with the trend of CCC for different cases such as when CCC is positive or negative.

Coupled autoregressive (AR) processes were generated as per Eq. (15). Also, linearly coupled tent maps were generated as per Eqs. (17) and (18). Symmetric KL and JSD between distribution P and Q of coupled processes are estimated as per Eqs. (12) and (14) respectively. (12) DSymm KLP,Q=DKLP∥Q+DKLQ∥P,

where, (13) DKLP∥Q= ∑iPilogPiQi,DKLQ∥P= ∑iQilogQiPi.

(14) JSDP∥Q=12DP∥M+12DQ∥M,

where, M=12P+Q. KL and JSD values are in unit of nats.

Curves for KL, JSD and CCC estimated for increasing coupling between AR processes of order 1 and linearly coupled tent maps are shown in Figs. 1 and 2 respectively. Results for non-linear coupling of tent maps are similar to that for linear coupling and are included (Fig. S10, Section S4.1). The values displayed represent the mean over 50 trials. As the degree of coupling is varied for AR processes, there is no clear pattern in KL and JSD values. CCC values increase in the positive direction as expected for increasing coupling, signifying that the dynamical influence from Y to X is similar to the influence on X from its own past. Also, when we took larger number of trials for AR, the values obtained by KL and JSD become confined to a smaller range and seem to converge towards a constant value indicating that the distributions of X and Y are quite similar. However, in case of coupled tent maps (both linear and non-linear coupling), as coupling is increased, the divergence between the distributions of the two coupled processes increases, indicating that their distributions are becoming very different. The values of CCC grow in the negative direction showing that with increasing coupling the independent process Y has a very different dynamical influence on X compared to X’s own past. Subsequently, due to the synchronization of Y and X, KL, JSD as well as CCC become zero. With these graphs, it may not be possible to find a universal threshold for the absolute values of KL/JSD above which CCC will show negative sign. However, if the distributions of the two coupled processes exhibit an increasing divergence (when the coupling parameter is varied) then it does indicate that the independent process would have a very different dynamical influence on the dependent one when compared with that of the dependent process’ own past, suggesting that the value of CCC will grow in the negative direction. The fact that KL/JSD and CCC do not have a one-to-one correspondence is because the former (KL and JSD) operate on first order distributions while the latter (CCC) is able to capture higher-order dynamical influences between the coupled processes. For non-stationary processes, our measure would still be able to capture the kind of dynamical influence, though distributions are not static.

Figure 1 Mean values of divergence between distributions of coupled AR(1) processes using Symmetric Kullback–Leibler (KL) (A) and Jensen Shannon (JSD) divergences (in nats) (B), and the mean causality values estimated using CCC from Y to X (solid line-circles, black) and X to Y (solid line-crosses, magenta), as the degree of coupling, ϵ is varied (C).

CCC values increase with increasing ϵ. There is no similarity in the trend of KL/JSD to CCC.

Figure 2 Mean values of divergence between distributions of linearly coupled tent maps using Symmetric Kullback Leibler (KL) (A) and Jensen Shannon (JSD) divergences (in nats) (B), and the mean causality values estimated using CCC from Y to X (solid line-circles, black) and X to Y (solid line-crosses, magenta) (C), as the degree of coupling, ϵ is varied.

For ϵ < 0.5, CCC and KL/JSD are highly negatively correlated.

Both positive and negative CCC imply significant causal influence (CCC≈0 implies either no causal influence or identical processes), but the nature of the dynamical influence of the cause on the effect is very different in these two cases. Causality turning ‘negative’ does not seem very intuitive at first, but all that it signifies is that the past of the cause variable makes the dynamics of the effect variable less predictable than its (effect’s) own past. Such a unique feature could be very useful for real world applications in terms of ‘controlling’ the dynamics of a variable being effected by several variables. If a particular cause, out of several causes that makes the caused ‘less predictable’ and has ‘intrinsically different’ dynamics from that of the effect, needs to be determined and eliminated, it can be readily identified by observing the sign of CCC. Informed attempts to inhibit and enforce certain variables of the system can then be made.

As the existing model-free methods of causality can extract only ‘associational causality’ and ignore the influence that the cause has on dynamics of the caused, it is impossible for them to comment on the nature of this dynamical influence, something that CCC is uniquely able to accomplish. Obviously, model based methods give full-fledged information about ‘the kind of dynamical influence’ owing to the model equations assumed. However, if there are no equations assumed (or known), then the sign and magnitude of CCC seems to be the best choice to capture the cause–effect relationship with additional information on the similarity (or its lack of) between the two dynamics.

Results and Discussion

A measure of causality, to be robust for real data, needs to perform well in the presence of noise, filtering, low temporal and amplitude resolution, non-uniformly sampled signals, short length time series as well as the presence of other causal variables in the system. In this section, we rigorously simulate these cases and evaluate the performance of CCC measure by comparing with existing measures—Granger Causality (GC) and Transfer Entropy (TE). Owing to space constraints, some of these results are included in Section S4. In the last sub-section, we test CCC on real-world datasets. In all cases, we take the averaged value of CCC over entire time series as computed by Eq. (8) (or Eq. (11) in the conditional case) and the parameters for CCC estimation are chosen as per the selection criteria and rationale discussed in Section S3. GC estimation is done using the MVGC toolbox (Barnett & Seth, 2014) in its default settings and TE estimation is done using MuTE toolbox (Montalto, Faes & Marinazzo, 2014). Akaike Information Criteria is used for model order estimation with the maximum model order set to 20 in the MVGC toolbox, except where specified. The maximum number of lags to take for autocorrelation computation is done automatically by the toolbox. In the MuTE toolbox, the approach of Non Uniform Embedding for representation of the history of the observed processes and of Nearest Neighbor estimator for estimating the probability density functions is used for all results in this paper. The number of lags to consider for observed processes was set to 5 and the maximum number of nearest neighbors to consider was set to 10.

Varying unidirectional coupling

AR(1)

Autoregressive processes of order one (AR(1)) were simulated as follows. X and Y are the dependent and independent processes respectively.

(15) Xt=aXt−1+ϵYt−1+εX,t

Yt=bYt−1+εY,t,

where a = 0.9, b = 0.8, t = 1 to 1,000s, sampling period = 1s. ϵ is varied from 0 − 0.9 in steps of 0.1. Noise terms, εY, εX = νη, where ν = noise intensity = 0.03 and η follows standard normal distribution. Figure 3 shows the performance of CCC along with that of TE and GC as mean values over 50 trials, (CCC settings: L = 150, w = 15, δ = 80, B = 2). Standard deviation of CCC, TE and GC values are shown in Fig. 4.

Figure 3 Mean causality values estimated using CCC (A), TE (B) and GC (C) for coupled AR(1) processes, from Y to X (solid line-circles, black) and X to Y (solid line-crosses, magenta) as the degree of coupling, ϵ is varied.

CCC, TE as well as GC are able to correctly quantify causality.

Figure 4 Standard deviation of causality values estimated using CCC (A), TE (B) and GC (C) for coupled AR(1) processes, from Y to X (solid line-circles, black) and X to Y (solid line-crosses, magenta) as the degree of coupling, ϵ is varied.

With increasing coupling, the causality estimated by CCC, TE as well as GC increases.

AR(100)

Autoregressive processes of order hundred (AR(100): X dependent, Y independent) were simulated as follows. (16) Xt=aXt−1+ϵYt−100+εX,tYt=bYt−1+εY,t,

where a = 0.9, b = 0.8, t = 1 to 1,000s, sampling period = 1s. ϵ is varied from 0 − 0.9 in steps of 0.1. Noise terms, εY, εX = νη, where ν = noise intensity = 0.03 and η follows standard normal distribution. Figure 5 shows the performance of CCC along with that of TE and GC, as mean values over 50 trials (CCC settings: L = 150, w = 15, δ = 80, B = 2). Maximum model order was set to 110 in the MVGC toolbox.

Figure 5 Mean causality values estimated using CCC (A), TE (B) and GC (C) for coupled AR(100) processes, from Y to X (solid line-circles, black) and X to Y (solid line-crosses, magenta) as the degree of coupling, ϵ is varied.

Only CCC is able to reliably estimate the correct causal relationship for all values of ϵ while TE and GC fail.

CCC values increase steadily with increasing coupling for the correct direction of causation. TE fails as it shows higher causality from X to Y for all ϵ. GC also shows confounding of causality values in two directions. Thus, causality in coupled AR processes with long-range memory can be reliably estimated using CCC and not using TE or GC. Range of standard deviation of CCC values from Y to X is 0.0076 to 0.0221 for varying parameter ϵ and that from X to Y is 0.0039 to 0.0053. These values are much smaller than the mean CCC estimates and thus, causality estimated in the direction of causation and opposite to it remain well separable. For TE, Y to X, standard deviation range is 0.0061 to 0.0090 and X to Y, standard deviation range is 0.0082 to 0.0118. For GC, Y to X, standard deviation range is 0.0012 to 0.0033 and X to Y, standard deviation range is 0.0015 to 0.0034.

Tent map

Linearly coupled tent maps were simulated as per the following equations. Independent process, Y, is generated as:

(17) Yt=2Yt−1,0≤Yt−1<1∕2,

Yt=2−2Yt−1,1∕2≤Yt−1≤1.

The linearly coupled dependent process, X, is as below:

(18) Xt=ϵYt+1−ϵht,

ht=2Xt−1,0≤Xt−1<1∕2,

ht=2−2Xt−1,1∕2≤Xt−1≤1,

where ϵ is the degree of linear coupling.

The length of the signals simulated in this case was 3,000, i.e., t = 1 to 3,000s, sampling period = 1s and the first 2,000 transients were removed to yield 1,000 points for causality estimation. Figure 6 shows the performance of CCC and TE for linearly coupled tent maps as ϵ is varied (CCC settings: L = 100, w = 15, δ = 80, B = 8). CCC and TE comparison was also done for increasing coupling in the case of non-linearly coupled tent maps. These results are included in the Section S4.1. Results obtained are similar to the linear coupling case. The assumption of a linear model for estimation of GC was proved to be erroneous for most trials and hence GC values are not displayed. As ϵ is increased for both linear and non-linear coupling, TEY→X increases in the positive direction and then falls to zero when the two series become completely synchronized at ϵ = 0.5. The trend of the magnitude of CCC values is similar to TE, however, CCCY→X increment is in the negative direction. This is because of the fact that with increasing coupling the kind of dynamical influence from Y to X becomes increasingly different than the dynamical influence from the past values of X to itself.

Figure 6 Mean of causality values estimated using CCC (A) and TE (B) for linearly coupled tent maps, from Y to X (solid line-circles, black) and X to Y (solid line-crosses, magenta) as the degree of coupling is increased.

With increasing coupling (until synchronization), magnitude of CCC and TE values increases. CCC values are negative while TE are positive.

In case of linear coupling, range of standard deviation of CCC values from Y to X is 0.0050 to 0.0087 for different values of ϵ and that from X to Y is 0.0051 to 0.0100. For TE, Y to X, standard deviation range is 0 to 1.4851 and X to Y, standard deviation range is 0 to 1.4225. For non-linear coupling, the range of standard deviation values are included in Section S4.1.

For both CCC and TE, standard deviation values obtained indicate that there might be confounding in the causality values in the direction of causation and the direction opposite to causation for low values of ϵ.

Varying process noise

The performance of measures as process noise is varied is shown in Fig. 7 for coupled AR processes simulated as in Eq. (15), where a = 0.9, b = 0.8, ϵ = 0.8, t = 1 to 1,000s, sampling period = 1s, number of trials = 50. Noise terms, εY, εX = νη, where ν = noise intensity, is varied from 0.01 to 0.1 and η follows standard normal distribution. CCC settings: L = 150, w = 15, δ = 80, B = 2. The range of standard deviation of CCC values from Y to X is 0.0162 to 0.0223 for different values of ϵ and that from X to Y is 0.0038 to 0.0058. For TE, Y to X, standard deviation range is 0.0182 to 0.0267 and X to Y, standard deviation range is 0.0063 to 0.0104. For GC, Y to X, standard deviation range is 0.0314 to 0.0569 and X to Y, standard deviation range is 0.0001 to 0.0002.

Figure 7 Mean causality values estimated using CCC (A), TE (B) and GC (C) for coupled AR processes, from Y to X (solid line-circles, black) and X to Y (solid line-crosses, magenta) as the intensity of noise, ν is varied.

All the three measures perform well in this case.

The performance of all three measures is fairly good in this case. Only GC values show a slightly increasing trend with increasing noise intensity.

Non uniform sampling

Results for causality testing on uniformly downsampled signals are included in the Section S4.2. Non-uniformly sampled/non-synchronous measurements are common in real-world physiological data acquisition due to jitters/motion-artifacts as well as due to the inherent nature of signals such as heart rate signals (Laguna, Moody & Mark, 1998). Also, in economics, the case of missing data is common (Baumöhl & Vỳrost, 2010). To realistically simulate such a scenario, non-uniform sampling was introduced by eliminating data from random locations of the dependent time series and then presenting the resulting series as a set with no knowledge of the time-stamps of the missing data. The percentage of non-uniform sampling/non-synchronous measurements (α) is the percentage of these missing data points.

AR processes with non-uniformly sampled signals were simulated as per Eq. (15) with b = 0.7, a = 0.9, ϵ = 0.8. Noise terms, εY, εX = νη, where ν = noise intensity = 0.03 and η follows standard normal distribution. Length of original time series, N = 2, 000, and is reduced upon increasing the percentage non-uniform sampling α. In order to match the lengths of the two time series, Y, the independent time series, is appropriately truncated to match the length of the dependent signal, X (this results in non-synchronous pair of measurements). CCC settings used: L = 150, w = 15, δ = 80, B = 2. Mean causality estimated for 10 trials using the three measures with increasing α, while ν = 0.03, are shown in Fig. 8.

Figure 8 Mean causality values estimated using CCC (A), TE (B) and GC (C) for coupled AR processes from Y to X (solid line-circles, black) and X to Y (solid line-crosses, magenta) as the percentage of non-uniform sampling α is varied.

CCC is the only measure that shows reliable, consistent and correct estimates of causality.

Linearly coupled tent maps with non-uniformly sampled signals were simulated as per Eqs. (17) and (18) with ϵ = 0.3. Length of original time series, N = 2000, and is reduced upon increasing the percentage non-uniform sampling α. In order to match the lengths of the two time series, Y, the independent time series, is appropriately truncated to match the length of the dependent signal, X (this results in non-synchronous pair of measurements). CCC settings used: L = 100, w = 15, δ = 80, B = 8. Mean causality estimated for 10 trials using the three measures with increasing increasing α, while ν = 0.03, are shown in Fig. 9.

Figure 9 Mean causality values estimated using CCC (A), TE (B) and GC (C) for coupled tent maps from Y to X (solid line-circles, black) and X to Y (solid line-crosses, magenta) as the percentage of non-uniform sampling is varied.

CCC is able to distinguish the causality direction but the separation between values is small. TE and GC completely fail.

As the results clearly indicate, both TE and GC fail when applied to non-uniformly sampled coupled AR and tent map processes. CCC values are relatively invariant to non-uniform sampling and thus could be employed in such scenarios.

Filtering of coupled signals

Acquired data preprocessing often involves low pass filtering to smooth out the signal (Teplan, 2002). At other times, high pass filtering is required to remove low frequency glitches from a high frequency signal. Also, when the signals acquired are sampled at low frequencies, the effects due to decimation and filtering may add up and result in poorer estimates of causality. This is often the case in fMRI signals (Glover, 2011; Kim, Richter & Uurbil, 1997).

To test these scenarios, AR processes were simulated as below: (19) Yt=0.7Yt−5+εY,t,Xt=0.9Xt−5+0.8Yt−1+εX,t,

where, noise terms, εY, εX = νη, where ν = noise intensity = 0.03 and η follows standard normal distribution.

Causality values were estimated using CCC, TE and GC when simulated signals are low pass filtered using a moving average window of length 3 with step size 1. The results are shown in Table 1 as mean values over 10 trials. CCC settings used: L = 150, w = 15, δ = 80, B = 2. The performance of the measures when coupled signals are decimated to half the sampling rate and then low pass filtered are also included in the table. The length of the original signal simulated is 2000 and is reduced to 1998 upon filtering and to 998 upon filtering and decimation.

Table 1 Mean CCC, TE and GC estimates for coupled AR processes Y (independent) and X (dependent) as it is, upon filtering and upon decimation and filtering.

System	CCC	TE	GC	
	Y → X	X → Y	Y → X	X → Y	Y → X	X → Y	
Original	0.0908	−0.0041	0.2890	0.0040	0.3776	0.0104	
Filtered	0.0988	0.0018	0.2398	0.0170	0.4787	0.0056	
Decimated and filtered	0.0753	0.0059	0.1270	0.0114	0.4321	0.0596	

From the table, we see that CCC can distinguish the direction of causality in the original case as well as in the filtering and decimation plus filtering case. Erroneously, TE shows significant causality in the direction opposite to causation upon filtering as well as upon decimation and filtering and GC shows significant causality in the direction opposite to causation upon decimation and filtering. By this we can infer that CCC is highly suitable for practical applications which involve pre-processing such as filtering and decimation of measurements.

Conditional CCC on short length MVAR system

A system of three variables was simulated as per the following equations— (20) Zt=0.8Zt−1+ϵZ,t,Xt=0.9Xt−1+0.4Zt−100+ϵX,t,Yt=0.9Yt−1+0.8Zt−100+ϵY,t,

where the noise terms, εZ, εX, εY = νη, ν = noise intensity = 0.03 and η follows standard normal distribution. Length of time series simulated was 300 and first 50 transients were removed to yield short length signals of 250 time points.

The coupling direction and strength between variables X, Y, Z are shown in Fig. 10A. The mean values of causality estimated over 10 trials using CCC, TE and GC are shown in Fig. 10 tables, (b), (c) and (d) respectively. CCC settings used: L = 150, w = 15, δ = 20, B = 2. In the tables, true positives are in green, true negatives in black, false positives in red and false negatives in yellow. CCC detects correctly the true positives and negatives. GC, detects the true positives but also shows some false positive couplings. TE, performs very poorly, falsely detecting negatives where coupling is present and also showing false positives where there is no coupling.

Figure 10 Mean causality values estimated using CCC (B), TE (C) and GC (D) for a system of three AR variables coupled as in (A).

True positives are in green, true negatives in black, false positives in red and false negatives in yellow.

Real Data

CCC was applied to estimate causality on measurements from two real-world systems and compared with TE. System (a) comprised of short time series for dynamics of a complex ecosystem, with 71 point recording of predator (Didinium) and prey (Paramecium) populations, reported in Veilleux (1976) and originally acquired for Jost & Ellner (2000), with first 9 points from each series removed to eliminate transients (Fig. 11A). Length of signal on which causality is computed, N = 62, CCC settings used: L = 40, w = 15, δ = 4, B = 8. CCC is seen to aptly capture the higher (and direct) causal influence from predator to prey population and lower influence in the opposite direction (see Fig. 11). The latter is expected, owing to the indirect effect of the change in prey population on predator. CCC results are in line with that obtained using Convergent Cross Mapping (Sugihara et al., 2012). TE, on the other hand, fails to capture the correct causality direction.

Figure 11 CCC, TE on real-world time series. (A) Time series showing population of Didinium nasutum (Dn) and Paramecium aurelia (Pn) as reported in Veilleux (1976), (B) Stimulus current (I) and voltage measurements (V) as recorded from a Squid Giant Axon (‘a3t01’) in Paydarfar, Forger & Clay (2006). (C): Table showing CCC and TE values as estimated for systems (A) and (B).

System (b) comprised of raw single-unit neuronal membrane potential recordings (V, in 10V) of squid giant axon in response to stimulus current (I, in V, 1V = 5 µA/cm2), recorded in Paydarfar, Forger & Clay (2006) and made available by Goldberger et al. (2000). We test for the causation from I to V for three axons (1 trial each) labeled ‘a3t01’, ‘a5t01’ and ‘a7t01’, extracting 5,000 points from each recording. Length of signal on which causality is computed, N = 5, 000, CCC settings used: L = 75, w = 15, δ = 50, B = 2. We find that CCCI→V is less than or approximately equal to CCCV→I and both values are less than zero for the three axons (Fig. 11), indicating negative causality in both directions. This implies bidirectional dependence between I and V. Each brings a different dynamical influence on the other when compared to its own past. TE fails to give consistent results for the three axons.

Conclusions

In this work, we have proposed a novel data-based, model-free intervention approach to estimate causality for given time series. The Interventional Complexity Causality measure (or ICC) based on capturing causal influences from the dynamical complexities of data is formalized as Compression-Complexity Causality (CCC) and is shown to have the following strengths—

• CCC operates on windows of the input time series (or measurements) instead of individual samples. It does not make any assumption of the separability of cause and effect samples.

• CCC doesn’t make any assumptions of stochasticity, determinism, gaussianity, stationarity, linearity or markovian property. Thus, CCC is applicable even on non-stationary/ non-linear/non-gaussian/non-markovian, short-term and long-term memory processes, as well as chaotic processes. CCC characterizes causal relationship based on dynamical complexity computed from windows of the input data.

• CCC is uniquely and distinctly novel in its approach since it does not estimate ‘associational’ causality (first rung on Ladder of Causation) but performs ‘intervention’ (second rung on the Ladder of Causation) to capture causal influences from the dynamics of the data.

• The point of ‘intervention’ (length L for creating the hypothetical data: Ypast + ΔX) is dependent on the temporal scale at which causality exists within and between processes. It is determined adaptively based on the given data. This makes CCC a highly data-driven/data-adaptive method and thus suitable for a wide range of applications.

• Infotheoretic causality measures such as TE and others need to estimate joint probability densities which are very difficult to reliably estimate with short and noisy time series. On the other hand, CCC uses Effort-To-Compress (ETC) complexity measure over short windows to capture time-varying causality and it is well established in literature that ETC outperforms infotheoretic measures for short and noisy data (Nagaraj & Balasubramanian, 2017a; Balasubramanian & Nagaraj, 2016).

• CCC can be either positive or negative (unlike TE and GC). By this unique property, CCC gives information about the kind of causal influence that is brought by one time series on another, whether this influence is similar (CCC > 0) to or different (CCC < 0) from the influence that the series brings to its own present.

• Negative CCC could be used for ‘control’ of processes by intervening selectively on those variables which are dissimilar (CCC < 0)/similar (CCC > 0) in terms of their dynamics.

• CCC is highly robust and reliable, and overcomes the limitations of existing measures (GC and TE) in case of signals with long-term memory, low temporal resolution, noise, filtering, non-uniform sampling (non-synchronous measurements), finite length signals, presence of common driving variables as well as on real datasets.

We have rigorously demonstrated the performance of CCC in this work. Given the above listed novel properties of CCC and its unique model-free, data-driven, data-adaptive intervention-based approach to causal reasoning, it has the potential to be applied in a wide variety of real-world applications. Future work would involve testing the measure on simulated networks with complex interactions as well as more real world datasets. We would like to further explore the idea of negative CCC and check its relation to Lyaupnov exponent (for chaotic systems) which can characterize the degree of chaos in a system. It is also worthwhile to explore the performance of other complexity measures such as Lempel–Ziv complexity for the proposed Interventional Complexity Causality.

We provide free open access to the CCC MATLAB toolbox developed as a part of this work. See Section S5 for details.

Supplemental Information

Text S1 Details regarding proposed Compression-Complexity Causality (CCC) method that are not included in the main article

This file provides explanation of how dictionary construction is done for estimating conditional CCC for multi-variate measurements. A table for detailed description of how CCC values can be computed to be either positive or negative is included here. We also describe the criteria and rationale for choosing the parameters of CCC and details of our MATLAB implementation that is made available for free download and use. Additional results of testing of CCC on simulations which could not be accommodated in the main paper are included here.

Click here for additional data file.

Dataset S1 MATLAB toolbox for Compression-Complexity Causality estimation

For description of the toolbox demo files and functions, please refer to Section 6 of the supplementary text and the readme file included in the toolbox.

Click here for additional data file.

Aditi Kathpalia is thankful to Manipal Academy of Higher Education for permitting this research as part of the PhD programme.

List of abbreviations

AR Autoregressive

C(⋅ ) Complexity

CC Dynamical Compression-Complexity

CCC Compression-Complexity Causality

CR Complexity Rate

ETC(⋅ ) Effort-to-Compress

GC Granger Causality

JSD Jensen–Shannon Divergence

LZ Lempel–Ziv Complexity

MVAR Multivariate Autoregressive

C(⋅, ⋅ ) Joint Complexity

CC¯ Averaged Dynamical Compression-Complexity

CCC¯ Averaged Compression-Complexity Causality

DC Dynamical Complexity

ETC(⋅, ⋅ ) Joint Effort-to-Compress

ICC Interventional Complexity Causality

KL Kullback–Leibler Divergence

TE Transfer Entropy

Additional Information and Declarations

Competing Interests

Author Contributions

Data Availability

1 Henceforth, the same variables are used to denote the binned/encoded versions of the blocks.

2 It should be mentioned that, strictly speaking, KL and JSD are not distance measures since they don’t satisfy the triangle inequality.

The authors declare there are no competing interests.

Aditi Kathpalia and Nithin Nagaraj conceived and designed the experiments, performed the experiments, analyzed the data, contributed reagents/materials/analysis tools, prepared figures and/or tables, performed the computation work, authored or reviewed drafts of the paper, approved the final draft.

The following information was supplied regarding data availability:

In Text S1, we provide details of our proposed method Compression-Complexity Causality (CCC). The text also provides details of the MATLAB toolbox for computation of CCC, made available as Supplemental Material.

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
