# Peer review of "Data-based intervention approach for Complexity-Causality measure"

_PeerJ Computer Science, doi:10.7717/peerj-cs.196_

## Round 0.1 · original submission · Major Revisions

I encourage the Authors to carry out a major revision of the manuscript that takes into account the observations of the reviewers, with special attention to addressing the major observations of Reviewer #2, which I consider critically important to consider acceptance of the manuscript.

Reviewer 1 ·

Basic reporting

Dear authors
thanks for submitting this interesting paper.
The results seem promising, the description needs some smoothing and some extra info, and it would be good to abandon the too often used approach of "here's a good method that outperforms other", since this is rarely the case, and the field would benefit from a more global view in which convenient aspects of several approaches are merged.


You mention that Granger causality is a "model free" approach, which it isn't. Transfer Entropy is. GC is rooted on autoregressive models. Maybe you meant "data driven".

The introduction of the paper is a bit confusing. I don't think that these methods address any of the ladders of causality, and maybe they shouldn't either. Of course when we apply these methods to real data, we are in a causal framework in the sense of the counterfactuals, but again, I find the introductory sentences misleading and dissociated from the actual content of the paper.

Experimental design

You could maybe better explain the philosophy behind your method, and whether it addresses the effects or the mechanism. Also the applications are either abstract benchmarks (still useful) or a case in which a ground truth or intervention is difficult to assess.

Validity of the findings

While your approach seems powerful for the data presented, it would be good and fair to state possible scenarios in which the performance is not so good.

I understand that Granger Causality and Transfer Entropy are among the most used approach, but other measures exist, such as the Convergent Cross Mapping http://science.sciencemag.org/content/338/6106/496.

Also, other approaches have proposed compression as a tool to evaluate causality, see for example

https://arxiv.org/abs/1611.00261

https://eda.mmci.uni-saarland.de/pubs/2016/origo-budhathoki,vreeken.pdf

Could you maybe comment on these aspects?

Reviewer 2 ·

Basic reporting

The paper "Data based intervention approach for Complexity-Causality measure" proposes a new model-independent measure aimed at describing high-order causal relations between signals. The authors provide context for the relevance of the proposed measure, comparing it against traditional model-free measures and show that their measure is generally able to provide better discrimination in simple synthetic cases and a couple of real systems.
I commend the authors for their extensive work explaining the new measure with different examples of growing complexity. I highlight a weakness in my comments below regarding the lack of comparison with a model-dependent case. The manuscript is written clearly, although occasionally there is a missing preposition or article. I recommend a further thorough check of the manuscript. Moreover, code is provided to replicate the analysis.
Overall, I think the manuscript is a valid contribution. I however harbor some doubts about the capacity and usefulness of the new measure in complex networked systems, which are the systems where most likely model-based measures fail or are unavailable. Hence, as mentioned in the comments below, I think that, before being accepted, the paper should include the comparison with a case where model-dependent is available, even if for a simple system.

Experimental design

no comment

Validity of the findings

Major comments:
- the authors throughout the paper compare first-order with high-order AR models in order to show that, while TE/GC sometimes work for AR(1) models, they consistently fail for higher-order one, thus highlighting the capacity of CCC to capture hierarchically higher causality effects. This is good, but what I would like to see is whether CCC is able to outperform specific model-dependent measures and observations in systems where we do have such specific equations.
Indeed, the authors mention DCM and SEM too: it would be great if it were possible to show that the direction and intensity of interactions picked up by DCM in real-data (even in one of the examples with few-nodes in Friston's DCM toolbox on EEG or fMRI data) corresponds to that captured by CCC. I understand that the authors mention something similar as future work, but given their claims about CCC, I would really like to see this in the paper.

- Following up, how hard computationally is to compute CCC versus existing measures of complexity? Is this a tool that can be thrown at data cheaply or is it instead to heavy to use as a first general-purpose step of analysis? Can we imagine to scale this up to the case of hundreds of signals, as is typically the case in neuroimaging applications?

Additional comments

- There are no error bars reported anywhere in the plots. Assuming that the curves shown are obtained over repeated simulations of the processes for each parameter choice, it would be interesting to understand the variance/magnitude of fluctuations around the reported mean values for both GC, TE and CCC.

- the figure labels are acceptable, but a little small; I'd recommend making the labels larger for improved readability.

---

## Round 0.2 · Minor Revisions

Based on combined assessment of the two Reviewers who read the original manuscript, and of an additional Reviewer, I'm glad to see that the manuscript has significantly improved.

The issue raised by Reviewer 1 regarding the comparison to DCM, however, needs to be addressed by either 1) dropping the DCM comparison in a minor revision of the manuscript, or 2) carrying out a major revision where the DCM comparison is carried out more exhaustively (see comments by Reviewer 1 to this end). I don't think that this would fit well the scope of the current manuscript, so I would recommend to proceed as in 1) and drop the DCM comparison. Hence my decision to ask for a minor revision.

In the context of this new revision, I would kindly ask the Authors to also compulsorily address as many of the comments by Reviewer 3 as possible.

Reviewer 1 ·

Basic reporting

nothing to add

Experimental design

There are many EEG and fMRI data available, openneuro.org has hundreds of them, not to mention the benchmark EEG and fMRI data included in the SPM package. So a comparison with DCM is certainly feasible. On the other hand we should not put all connectivity estimates in the same box, DCM is a confirmatory approach meant to disclose the involvment of synaptic parameters, and should not be used for network discovery and to assess the simple presence of links.
A benchmark should be carefully chosen in order to map the effects of an intervention in the two cases.

Validity of the findings

The findings are valid, but I see the risk that all possible connectivity (or "causality") estimates are conflated, and in this case any comparison and interpretation is meaningless.

Reviewer 2 ·

Basic reporting

The authors improved significantly the manuscript according to my previous and the other referees' suggestions. In particular, they showed the effectiveness of their measure in specific cases of biological relevance.

Experimental design

no comment.

Validity of the findings

no comments

Reviewer 3 ·

Basic reporting

Overall the manuscript is clearly reported. A couple of minor issues...

The proposed method is based on the Effort-to-compress measure, but this is not described in the manuscript (only in the supplementary material). This severely effects the clarity of the article. This is particularly strange given that a number of other measures, which are not central to the proposed measure, are defined in the manuscript (e.g., KL-divergence and JS-divergence). I suggest including more details about the ETS in the main text.

The large number of acronyms make the manuscript bothersome to read. It would be useful to remove some of these or at least provides some reminders along the way. Also it seems that ICC and CCC are the same thing except that the latter is specifically using ETC as the notion of complexity.

Experimental design

The experimental design seems adequate.

Validity of the findings

The validity of the findings appear to be sound. However, I'd like to echo reviewer one's comment on stating the scenarios in which performance can be expected to be good and the scenarios for which it is expected to be bad. Perhaps a thorough analysis of this is beyond the scope of the current work. But without such analysis, the usefulness of the proposed approach may be limited as one can never be sure under what circumstances the method is applicable. Indeed this is a general limitation for so-called "model-free" approaches since the assumptions are ignored rather than clearly stated.

---

## Round 0.3 · accepted · Accept

The submitted minor revision satisfactorily addresses the remaining observation by the Reviewers.